# Identification of Cis-Regulatory Sequences Controlling Pollen-Specific Expression of Hydroxyproline-Rich Glycoprotein Genes in *Arabidopsis thaliana*

**DOI:** 10.3390/plants9121751

**Published:** 2020-12-10

**Authors:** Yichao Li, Maxwell Mullin, Yingnan Zhang, Frank Drews, Lonnie R. Welch, Allan M. Showalter

**Affiliations:** 1School of Electrical Engineering and Computer Science, Russ College of Engineering, Ohio University, Athens, OH 45701, USA; mm588613@ohio.edu (M.M.); yz209517@ohio.edu (Y.Z.); drews@ohio.edu (F.D.); 2Department of Environmental and Plant Biology, Molecular and Cellular Biology Program, College of Arts and Sciences, Ohio University, Athens, OH 45701, USA

**Keywords:** hydroxyproline-rich glycoproteins, cis-regulatory motifs, pollen-specific, machine learning, tissue-specific expression

## Abstract

Hydroxyproline-rich glycoproteins (HRGPs) are a superfamily of plant cell wall structural proteins that function in various aspects of plant growth and development, including pollen tube growth. We have previously characterized protein sequence signatures for three family members in the HRGP superfamily: the hyperglycosylated arabinogalactan-proteins (AGPs), the moderately glycosylated extensins (EXTs), and the lightly glycosylated proline-rich proteins (PRPs). However, the mechanism of pollen-specific HRGP gene expression remains unexplored. To this end, we developed an integrative analysis pipeline combining RNA-seq gene expression and promoter sequences to identify cis-regulatory motifs responsible for pollen-specific expression of HRGP genes in *Arabidopsis thaliana*. Specifically, we mined the public RNA-seq datasets and identified 13 pollen-specific HRGP genes. Ensemble motif discovery identified 15 conserved promoter elements between *A.*
*thaliana* and *A. lyrata*. Motif scanning revealed two pollen related transcription factors: GATA12 and brassinosteroid (BR) signaling pathway regulator BZR1. Finally, we performed a regression analysis and demonstrated that the 15 motifs provided a good model of HRGP gene expression in pollen (R = 0.61). In conclusion, we performed the first integrative analysis of cis-regulatory motifs in pollen-specific HRGP genes, revealing important insights into transcriptional regulation in pollen tissue.

## 1. Introduction

Tissue-specific gene expression patterns are maintained by the combinatorial binding of transcription factors (TFs) to DNA motifs in a cooperative and competitive manner. DNA motifs are specific short DNA sequences, often 8–20 nucleotides in length [1], which are statistically overrepresented in a given set of sequences. Extensive studies have been done to characterize regulatory factors and sequences responsible for tissue-specific gene expression in human and in mouse [2,3]. However, to our knowledge, there is no such study that elucidates transcriptional regulatory motifs responsible for pollen-specific gene expression, particularly for hydroxyproline-rich glycoproteins (HRGPs) in *Arabidopsis thaliana*.

Hydroxyproline-rich glycoproteins (HRGPs) are a superfamily of plant cell wall proteins involved in various aspects of plant growth and development [4]. The HRGP superfamily consists of three family members, the extensins (EXTs), arabinogalactan-proteins (AGPs), and proline-rich proteins (PRPs). Although all HRGPs contain hydroxyproline, the three family members are distinguished by their unique amino acid compositions, repeated amino acid motifs, and the degree and type of glycosylation. For example, AGPs can be identified by their biased amino acid compositions of Pro (P)/Hyp (O), Ala (A), Ser (S), and Thr (T); their frequent occurrence of AP and PA dipeptide repeats; and their large arabinose and galactose-rich polysaccharide chains attached to O residues. In contrast, EXTs tend to be rich in S, P/O, Val (V), Tyr (Y), Lys (K), and SOOOO pentapeptide repeats, and have multiple short arabinose oligosaccharide side chains attached to their O residues. Finally, PRPs are rich in P, V, K, Cys (C), and T; often have various P/O-rich amino acid repeat motifs in which not all the P residues are modified to form O; and are the least glycosylated members of the HRGP superfamily.

Bioinformatic programs analyzing genomic/proteomic data from the model genetic plant, *Arabidopsis thaliana*, have identified 166 HRGPs consisting of 85 AGPs, 59 EXTs, 18 PRPs, and 4 AGP/EXT hybrid HRGPs [4]. Most HRGP genes are widely expressed in a variety of plant organs and tissues, while others demonstrate more limited tissue-specific expression. Additionally, several HRGP genes are differentially expressed in response to particular biotic and abiotic stress conditions. This information has provided functional insight into the HRGP superfamily and is used by researchers to facilitate and guide further research in the field. However, one of the unexplored topics is the transcriptional regulation of HRGP genes, particularly in mature pollen (e.g., sperm cells).

To address this topic, we analyzed 113 RNA-seq data sets from Araport11 [5] and identified 13 pollen-specific HRGP genes, based on the tissue-specificity index (Tau [6]). Ensemble motif discovery was performed using Emotif-Alpha [7], resulting in the identification of 15 pollen-specific de novo motifs. Known motif matching based on PlantTFDB [8] and TOMTOM [9] identified interesting TFs that have been previously reported in pollen, such as GATA12 and BZR1. Regression analysis between HRGP gene expression and the identified motifs showed a significant correlation (R = 0.6, *p* < 0.01). Our results provide the first discovery of putative cis-regulatory elements in pollen-specific HRGP genes.

## 2. Results

### 2.1. Integrative Motif Discovery Pipeline for Pollen-Specific HRGPs

To answer the question of what cis-regulatory motifs control pollen-specific HRGP gene expression, we developed a systematic bioinformatic pipeline for interactive analysis of pollen-specific HRGP genes and de novo promoter motifs (Figure 1) based on several published databases and tools, including the gene expression database Araport11 [5]; the known transcription factor binding sites (TFBSs) database plantTFDB [8]; and the motif discovery tool Emotif-Alpha [7], which is an ensemble motif discovery pipeline that integrates 11 motif discovery tools, such as GimmeMotifs [10], DECOD [11], and DME [12]. We obtained 166 HRGP genes from [4], in which 13 genes were further defined as pollen-specific and 132 genes were defined as non-pollen (i.e., not expressed in pollen) based on public RNA-seq datasets. We then performed ensemble motif discovery using Emotif-Alpha and identified a set of 13 motifs (detected by filter A, a relaxed constraint) and a set of 3 motifs (detected by filter B, a rigorous constraint); we found one common motif in the two sets. Finally, using the results from filter A and filter B, we performed a regression analysis using the identified 15 motifs and the expression values of the 166 HRGP genes. The regression model showed that the identified motifs provided a good model of HRGP gene expression in pollen tissue (R = 0.61).

### 2.2. Identification of Pollen-Specific HRGP Genes

Tissue-specific gene expression analysis was performed using 113 RNA-seq samples in 11 different tissues from Araport11 [5]. We computed the tissue specificity index (Tau [6]) for each gene. Tau values vary from 0 to 1, where lower Tau values correspond to more universally expressed genes and higher Tau values correspond to genes that are expressed in a more tissue-specific manner. A gene is defined as tissue-specific if its Tau value is greater than 0.85. Thus, an HRGP gene is called pollen-specific if Tau > 0.85 and the highest expression level of the gene occurs in pollen tissue. Using these criteria, we have identified 13 pollen-specific HRGPs, including 8 EXTs and 5 AGPs (Table 1). To identify promoter motifs controlling pollen-specific HRGP gene expression, we further defined a background set of 132 non-pollen HRGP genes for discriminative motif discovery. A gene expression heatmap showed that the pollen-specific HRGPs were almost exclusively expressed in pollen, and non-pollen HRGPs were expressed in other tissues but not in pollen (Figure 2). We noticed that 12 pollen-specific HRGPs have been previously reported to be pollen-specific by an analysis of gene expression microarrays in [4], and they are known to play important functions in pollen. Specifically, PERK4, PERK5, PERK6, PERK7, PERK11, and PERK12 are involved in pollen tube growth [13]. FLA3-RNAi transgenic plants show abnormal pollen grains with less viability [14]. PEX4 is also involved in pollen tube growth [15]. AGP50 (BCP1) is required for male fertility [16]. AGP6 and AGP11 are important to pollen grain development, and homozygous double mutants lead to abnormal pollen grains [17,18]. AGP23 is predicted to play an important role in microspore development and/or pollen tube growth [19]. Interestingly, AT1G54215 (EXT32) is a newly identified pollen-specific HRGP, discovered by analyzing the public RNA-seq datasets. Due to cross-hybridization and saturation of signals, microarrays have limited detection capability and high background noise. RNA-seq, on the other hand, is far more sensitive and precise [20], thus enabling us to identify one additional pollen-specific HRGP (when compared to [4]).

### 2.3. Integrative Analysis Filter A: A Relaxed Set of Pollen-Specific HRGP Motifs

Next, we performed ensemble motif discovery using Emotif-Alpha [7] on the 13 pollen-specific HRGP genes against the background of 132 non-pollen HRGP genes. Gene promoters (within 1kb upstream of the translation start site) were retrieved from Ensembl Biomart [21]. Emotif-Alpha integrated 11 motif discovery tools and led to the identification of 3519 motifs in total.

Filter A selected motifs present in 11 (85%) or more of the pollen-specific HRGP genes and present in at most 30 (23%) non-pollen HRGP genes. This yielded 13 motifs, which were also conserved in *Arabidopsis lyrata*. Table 2 shows the 13 identified motifs. Interestingly, we found that 4 out of the 13 motifs matched to known motifs based on analysis using TOMTOM [9] (*p*-value < 0.001).

It is worth noting that the motif gimme_105_Improbizer AACACACGTTTATTAGATGTTT occurs in all 13 pollen-specific HRGP genes and this motif is highly similar (*p*-value = 1.6 × 10^−6^) to the known BZR1 (Brassinazole Resistant 1) binding motif. Brassinosteroid (BR) is an important class of steroid hormones in plants that regulates gene expression and cell development [22,23,24]. BZR1 is a key transcription factor in the BR signaling pathway, where the binding of BR to a cell surface receptor kinase (BRI1) directly regulates the phosphorylation of BZR1, which then binds to the promoters of BR responsive genes. BR was first discovered in pollen, where it regulates cell elongation. Although it was later found in all tissues, its highest abundance was in pollen, seeds, and fruit [22]. Indeed, cell wall modification is reported to be one of the major functions targeted by the BR pathway [22]. The discovery of BZR1-like binding sites in the promoters of pollen-specific HRGP suggests that these HRGPs are likely to be regulated by the BR signaling pathway.

### 2.4. Integrative Analysis Filter B: A Rigorous Set of Pollen-Specific HRGP Motifs

Filter B represents a more rigorous approach than Filter A, where known pollen expressed TFs (i.e., log2 gene expression ≥1) and their binding motifs were used to filter motifs. First, 99 motifs were identified, based on the cognate transcription factors expressed (based on Araport11 [5]) in pollen and obtained from the plantTFDB [8]. Known motif similarity found that 1341 de novo motifs out of the total 3519 motifs were highly similar to these 99 motifs (*p* < 0.001). The set of motifs was then filtered by conservation and number of occurrences, with redundant motifs removed.

Interestingly, two motifs discovered by Filter B (DME_ACDGWGMYA and DME_ARRTCYKVRG) matched with GATA9 binding sites (Table 3). Since GATA9 is the closest homolog of GATA12; it is likely that GATA12 also binds to these motifs. Interestingly, previous study has found GATA12 is highly expressed in mature pollen grains but is diminished in germinated pollen grains and pollen tubes [25], suggesting that these two motifs might play regulatory roles in pollen-specific gene expression in *Arabidopsis thaliana*. Additionally, Filter B found Motif gimme_143_MEME_4_w12, which was also found by the filter A approach; this motif matches a known binding site for the TED protein.

### 2.5. Modeling HRGP Gene Expression in Pollen

To investigate how well our identified pollen-specific motifs can predict HRGP gene expression in pollen, we conducted a regression analysis using scikit-learn [26]. The 15 identified motifs were mapped to the 166 HRGP gene promoters, and FIMO [27] motif mapping *p*-values (negative log10 transformed) were used as features. A gradient boosting tree was trained and evaluated using three-fold cross validation [26]. We obtained a good correlation between true HRGP gene expression and predicted values (R = 0.61, *p* = 3.31 × 10^−18^), indicating that the 15 identified motifs may control HRGP gene expression in pollen (Figure 3). We then extracted feature importance from the trained model; the BZR1-like motif, gimme_105_Improbizer, was the third most important motif, suggesting again the pollen-specific HRGP genes might be regulated by the BR signaling pathway (Appendix A).

## 3. Discussion

Elucidating transcriptional regulatory mechanisms and transcription factors are fundamental and critical to understanding gene expression, which controls the growth and development of all living things. In this study, we undertook the first analysis to examine and illustrate transcriptional regulatory mechanisms of pollen-specific HRGP genes in *Arabidopsis thaliana*, which are important for pollen growth and development [13,14,15,16,17,18,19]. Specifically, we employed the tissue-specificity index to define 13 pollen-specific HRGP genes. Using two different filters, including both relaxed and rigorous criteria, we identified 15 pollen-specific motifs that were matched to several known motifs, including BZR1 and GATA12, which are known to be pollen-specific and play key roles in cell wall modification [22,25]. Moreover, regression analysis showed that the identified motifs can be used to predict HRGP gene expression in pollen. Together, these results shed light on the transcriptional regulatory mechanisms of pollen-specific HRGP genes.

As more plant genomes have been sequenced, genome annotation and related bioinformatics analysis pipelines have become critical for researchers to further understand plant biology. This integrative analysis pipeline for studying cis-regulatory motif effects on tissue-specific genes is generalizable to other species. HRGP genes have been characterized in many other species, including *Oryza sativa*, *Brassica rapa*, and *Populus trichocarpa* [28]. However, the extent and mechanisms associated with the transcriptional regulation of these genes remains virtually unknown. Future studies can focus on the cis-regulatory motif analysis of HRGP gene promoters throughout the entire plant kingdom, which is significant to identify both conserved and divergent regulatory elements.

The HRGP superfamily has a nested classification hierarchy. For example, while HRGPs are composed of EXTs, AGPs, and PRPs, the EXT family can be further divided into classical EXTs, short EXTs, leucine-rich repeat extensins (LRXs), and proline-rich extensin-like receptor kinases (PERKs) [28]. Fine-grained HRGP classification systems, such as the protein sequence signatures identified by Bio OHIO 2.0 [29], suggest different gene transcriptional regulation mechanisms in different HRGP subfamilies. Future studies can focus on the promoter analysis in different HRGP subfamilies.

Gene expression is controlled by the complex combinatorial binding of transcription factors and epigenetic regulations of chromatin accessibility, histone modifications, and DNA methylation. Although our identified pollen-specific motifs have provided a good model for predicting HRGP gene expression (R = 0.61), this gene expression model is not yet complete. To capture the remaining unexplained variance in this model, we need to gain deeper understanding of the promoter architectures. Future studies can incorporate motif distance features and gene networks. Multi-omics data, such as ATAC-seq and ChIP-seq, will also help to better understand the transcriptional regulatory landscape in pollen.

## 4. Materials and Methods

### 4.1. Characterization of Pollen-Specific HRGP Genes

A list of 33,602 *Arabidopsis thaliana* genes with TAIR IDs was downloaded from the TAIR website [30]. For each gene in the list, its expression profile in 113 RNA-seq experiments was retrieved from Araport11 using the python API *intermine.webservice* [5]. The description of the RNA-seq dataset can be found at the Araport11 website [5]. To determine pollen-specific expression, the tissue specificity index, Tau, was used. In a recent benchmarking comparison, Tau was found to be the most robust and biologically relevant method [20]. Tau varies from 0 to 1, where lower Tau means more universally expressed and higher Tau means more tissue specifically expressed. As recommended in [5], genes with Tau > 0.85 were considered tissue-specific. The tissue type was determined by the largest expressed tissue. All expression values were log-transformed before calculating Tau; values <1 were set to 0 after log transformation [20]. Using this method, we have characterized tissue-specific expression patterns for 26,500 genes. For the gene expression heatmap, gene expression z-scores were calculated by seaborn clustermap [31]; larger values indicate higher expression.

A list of 166 HRGPs was reported by Showalter et al. [4]. Pollen-specific HRGPs are defined as a list of HRGPs that are pollen-specific expressed. Non-pollen HRGPs are defined as a list of HRGPs that have no expression (expression value is 0 after log transformation) in pollen.

### 4.2. Promoter Retrieval and Ensemble Motif Discovery

Promoters of pollen-specific HRGPs and non-pollen HRGPs were retrieved from Ensembl Plant v35 Biomart [21] web interface using gene stable ID, Flank (Gene) Coding Region, and Upstream flank 1000 bp.

To identify regulatory motifs for pollen-specific HRGPs, Emotif-Alpha [7], an ensemble motif discovery pipeline, was utilized. The foreground promoter set was the list of 13 pollen-specific HRGPs. The background promoter set was the list of 132 non-pollen HRGPs. Emotif-alpha has integrated 11 motif discovery tools: GimmeMotifs [10], MEME [32], Weeder [33], BioProspector [34], AMD [35], Homer [36], GADEM [37], MDmodule [38], Improbizer [39], DECOD [11], and DME [12]. Motif length was set to be 6–16 nt. FIMO [21] was used for motif scanning. The discriminative power of the motifs was assessed by a random forest classifier using the scikit-learn package. Motif similarity was assessed by TOMTOM [9]; two motifs were considered to be similar if their TOMTOM [9] *p*-value was less than 0.001. For similar motifs, the motif with fewer occurrences in pollen-specific HRGPs was filtered out.

### 4.3. Conservation Analysis

Conservation analysis was performed using the method adopted by Roy et al. [40]. Orthologous information between *A. thaliana* and *A. lyrata* were retrieved from Ensembl Plant Biomart v39 [21]. CLUSTALW2 [41] was used to do multiple sequence alignment with gap open penalty of 10 and extension penalty of 0.1. A motif was defined as conserved if it occurred at the same position in the orthologous promoter alignment.

### 4.4. Machine Learning

The 15 identified motifs were scanned on the 166 HRGP promoters using FIMO [27] and the negative log10 motif scanning *p*-values were used as machine learning features for regression. The regression algorithm was implemented using scikit-learn GradientBoostingRegressor with parameters of subsample = 0.3, criterion = “mae”, min samples_split = 5, max depth = 1 [26]. The evaluation was performed using 3-fold cross-validation using the KFold function.

## Figures and Tables

**Figure 1 plants-09-01751-f001:**
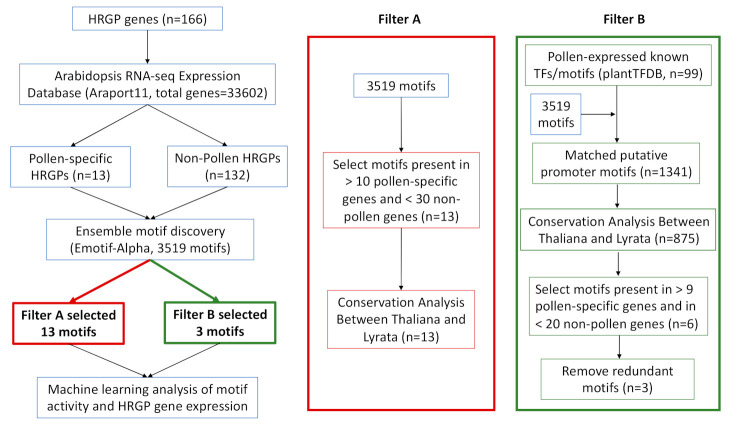
Workflow of an integrative analysis combining hydroxyproline-rich glycoproteins (HRGP) gene expression and promoter sequences. HRGP genes were split into pollen-specific and non-pollen (i.e., not expressed in pollen) based on tissue-specific gene expression analysis from Araport11 [5]. Ensemble motif discovery was performed on the aforementioned two sets and a total of 3519 motifs were identified, which were used by the two filters. Filter A is a more relaxed approach where the motifs are filtered based on the number of occurrences (>10 in pollen-specific HRGPs and <30 in non-pollen HRGPs) and conservation criterion. Filter B is a more rigorous approach, involving 99 motifs whose cognate transcription factors are expressed in pollen based on data obtained from plantTFDB [8]. Known motif similarity found that 1341 de novo motifs out of the total 3519 motifs were highly similar to the 99 motifs (*p*-value < 0.001). The set was then filtered by conservation and number of occurrences, with redundant motifs being removed. In total, filter A and filter B identified 15 motifs (with one motif being identified by both filter A and filter B) putatively controlling pollen-specific HRGP gene expression; these motifs were then fit into a regression model that integrated the promoter elements and HRGP gene expression.

**Figure 2 plants-09-01751-f002:**
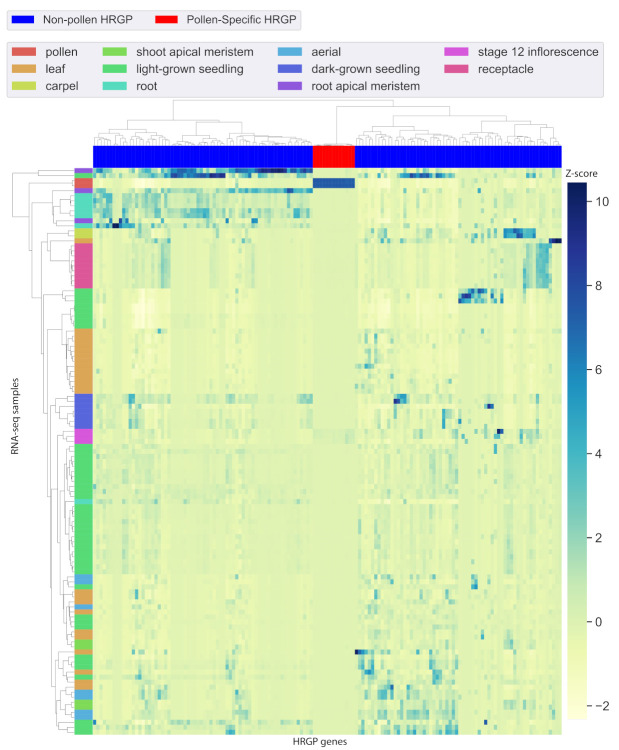
Gene expression heatmap showing pollen-specific HRGP gene expression. Rows represent the 113 RNA-seq datasets from 11 different tissues. Columns represent the 13 pollen-specific (red) and 132 non-pollen (blue) HRGP genes.

**Figure 3 plants-09-01751-f003:**
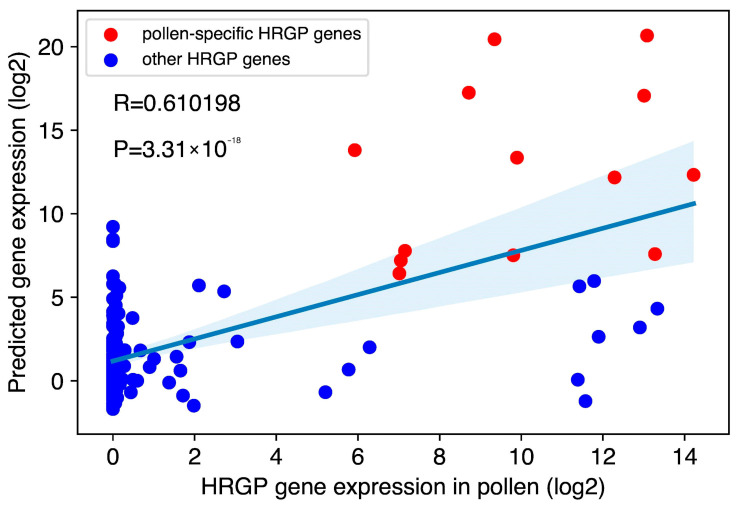
Regression analysis using the identified 15 pollen-specific motifs. Each point is an HRGP gene. Pollen-specific HRGP genes are highlighted in red. The X-axis is the mean HRGP gene expression in pollen and the Y axis is the predicted gene expression value. The Pearson correlation coefficient is 0.61 and the *p*-value is 3.31 × 10^−18^.

**Table 1 plants-09-01751-t001:** List of pollen-specific HRGP genes.

TAIR ID	Gene Name ^a^	Tissue Specificity Index ^b^	Expression in Pollen ^c^	Level of Expression ^d^	Reference ^e^
AT1G10620 *	PERK11	0.968	7.003	Extremely high	[13]
AT1G49270 *	PERK7	0.960	8.711	Extremely high	[13]
AT4G34440 *	PERK5	0.948	7.039	Extremely high	[13]
AT3G18810 *	PERK6	0.941	9.343	Extremely high	[13]
AT2G24450 *	FLA3	0.936	12.284	Extremely high	[14]
AT1G23540 *	PERK12	0.936	7.141	Extremely high	[13]
AT4G33970 *	PEX4	0.909	9.893	Extremely high	[15]
AT1G54215	EXT32	0.908	5.898	High	
AT2G18470 *	PERK4	0.880	9.803	Extremely high	[13]
AT1G24520 *	AGP50	0.879	13.274	Extremely high	[16]
AT3G01700 *	AGP11	0.872	13.079	Extremely high	[17,18]
AT5G14380 *	AGP6	0.862	13.008	Extremely high	[17,18]
AT3G57690 *	AGP23	0.856	14.220	Extremely high	[19]

* These genes have been reported to be pollen-specific in [4]. ^a^ Gene Name is adopted from [4], where some genes are renamed by the authors to indicate their protein sequence properties. ^b^ Tissue specificity index Tau is calculated using the formula presented in [6]. ^c^ Expression is represented using the median value after log 2 transformation. ^d^ Expression value is compared to number of standard deviations (stds) away from the mean value in all genes’ expression profile in pollen. Extremely high expressed genes are more than 3 *stds away from the mean and high expressed genes have are than 2 *stds but less than 3 *stds away from the mean. ^e^ References for known pollen-related functions.

**Table 2 plants-09-01751-t002:** Motifs identified by Filter A. The foreground coverage indicates the percentage (and number) of pollen-specific HRGP genes having an identified motif in its promoter region. The background coverage is the percentage (and number) of the non-pollen HRGP genes having that motif in its promoter region. The relative frequency is the foreground coverage divided by the background coverage. The matched transcription factor binding sites (TFBS) *p*-value is a known transcription factor binding site that relates to pollen along with the accompanying *p*-value.

Motif Name	Motif Logo	Foreground Coverage	Background Coverage	Best Matched TFBS (*p*-Value)
DME_GADGAYKAS	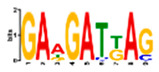	85% (11)	19% (25)	AT3G11280, MYB-LIKE PROTEIN (3.6 × 10^−4^)
DME_GATYTKRHG	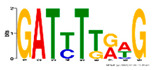	85% (11)	20% (27)	
DME_GRHTGDTGA	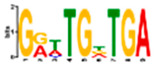	85% (11)	20% (27)	AT5G58620, TZF9 (1.3 × 10^−5^)
DME_MARKGDSRGA	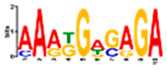	85% (11)	22% (29)	
gimme_102_Improbizer_GCGTTATACCCGAGGATCAG	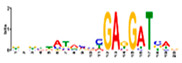	92% (12)	15% (20)	
gimme_104_Improbizer_GTGCAACGGAGAGT	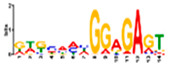	92% (12)	14% (18)	
gimme_105_Improbizer_AACACACGTTTATTAGATGTTT	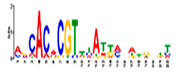	100% (13)	18% (24)	AT1G75080, BZR1 (1.6 × 10^−6^)
gimme_132_MEME_3_w10	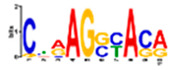	92% (12)	11% (15)	
gimme_13_BioProspector_w10_3	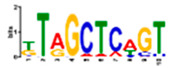	85% (11)	13% (17)	
gimme_143_MEME_4_w12	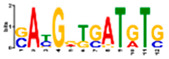	85% (11)	12% (16)	AT5G11260, TED5 (4.7 × 10^−4^)
gimme_146_MEME_7_w12	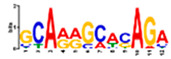	92% (12)	16% (21)	
gimme_16_BioProspector_w12_1	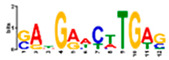	85% (11)	20% (27)	
gimme_92_MDmodule_Motif.12.3	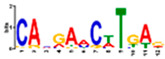	85% (11)	20% (26)	

**Table 3 plants-09-01751-t003:** Motifs identified in Filter B. The foreground coverage indicates the percentage (and number) of pollen-specific HRGP genes having the identified motif in their promoter regions. The background coverage is the percentage (and number) of the non-pollen HRGP genes having that motif in their promoter regions. The relative frequency is the foreground coverage divided by the background coverage. The matched TFBS *p*-value is a known transcription factor binding site that relates to pollen along with the accompanying *p*-value.

Motif Name	Motif Logo	Foreground Coverage	Background Coverage	Best Matched TFBS (*p*-Value)
DME_ACDGWGMYA	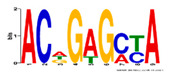	77% (10)	10% (13)	AT4G32890, GATA9 (5 × 10^−4^)
gimme_143_MEME_4_w12	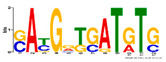	85% (11)	12% (16)	AT5G11260, TED5 (9 × 10^−4^)
DME_ARRTCYKVRG	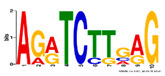	77% (10)	12% (16)	AT4G32890, GATA9 (1.8 × 10^−3^)

## Data Availability

All data and source code used in this study is available at https://github.com/YichaoOU/Pollen_specific_motifs.

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
