# Peer review of "Identification of Cis-Regulatory Sequences Controlling Pollen-Specific Expression of Hydroxyproline-Rich Glycoprotein Genes in Arabidopsis thaliana"

_plants, 2020, doi:10.3390/plants9121751_

Round 1

Reviewer 1 Report

The manuscript ‘Identification of Cis-Regulatory Sequences Controlling Pollen-Specific Expression of Hydroxyproline-Rich Glycoprotein Genes in Arabidopsis thaliana’ by Li et al., presents an in-depth high throughput analysis to identify the transcription regulators of HRGP genes explicitly in mature pollen.

I read this manuscript with great interest. I have a few major comments:

The manuscript is well written and presented.

In Table S1, the authors listed the details of 13 pollen-specific HRGP genes and mentioned that:-*These genes have been reported to be pollen-specific [Ref. 4]. In an in silico analysis, the authors identified the TFs that are predicted to regulate the expression of these 13 pollen-specific HRGP genes.  The TF binding sequences on these 13 genes' promoters are indicated for binding two TFs (GATA12 and BZR1).  The authors have rigorous statistical strength to their prediction; however, the manuscript lacks the physiological and molecular evidence about the regulation of HRGP genes by GATA12 and BZR1. The authors should validate if the expression of 13 HRGP genes alters in the pollens of gata12 and bzr1 mutants and whether the altered expression of HRGP genes in these mutants affects the pollen structure physiology (germination, pollen tube formation, etc.). These data will support the in silico predictions.      

Table S1 should be the main table, and the authors must provide the references for these genes where the pollen-related phenotype is known. For example:

AT3G01700- AGP11- which encodes an arabinogalactan protein expressed in pollen, pollen sac, and pollen tube. The agp11 mutant showed decreased fertility due to defects in pollen tube growth (Levitin et al., 2008; Coimbra et al., 2009).   

Line 175, the authors mentioned that 15 motifs were identified in 166 HRGP genes. These 15 identified motifs may control HRGP gene expression in pollen (Figure 3). The authors should provide details of these motifs.

The discussion reads fragmented and lacks physiological relevance.

Minor comments:

Please check Line 64 and 286

Line 215 for Arabidopsis thaliana (it should be in italic)

Reviewer 2 Report

All parts of the work are developed in a right way. Only for the discussion I suggest to improve it in order to detail better the possible future developments to detail a possible unique regulated gene expression model on pollen.

Reviewer 3 Report

The study was undertaken to reveal the cis-regulatory sequence for pollen-specific expression of Hydroxyproline-rich glycoproteins (HRGPs) by integrated analysis of RNA sequence gene expression and Promoter sequences. Results and conclusion of the study are presented well, however, there are concerns, which can be substantiated

  1. Even though the authors claim as the first report of HRGPs, the cis-regulatory sequence for pollen-specific, this report shows an average novelty/originality. There were many reports on the identification of protein sequences.
  2. This report will also have more advantages if the author explores this method to identify the regulatory sequence of HRGPs, in non-model crops.
  3. It is highly encouraged, authors can do experimental validation of the predicted sequences using the agrobacterium infiltration method
  4. It is also important to do the Arabidopsis genetic transformation studies,  It also is nice, if authors able to find the mutant with deletion of those sequences and doing transformation complementary using the cis-regulatory sequence of HRGPs.
  5.   if possible, authors are encouraged to do the gene knockout study in the Arabidopsis to validate. 

Round 2

Reviewer 1 Report

The authors have reasoned that due to COVID and short manuscript turnover period is not sufficient for validating if the expression of 13 HRGP genes alters in the pollens of gata12 and bzr1 mutants and whether the altered expression of HRGP genes in these mutants affects the pollen structure physiology (germination, pollen tube formation, etc.).

The current pandemic situation really hampered the lab works. It is acceptable that the mutant analysis for the pollen structure physiology (germination, pollen tube formation, etc.) will be challenging during the COVID situation. 

Perhaps, the authors could find a way to strengthen their predictions, possibly looking for reported gata12 and brz1 transcriptome to check the expression of 13 HRGPs. However, the authors should not use the short 10-day revision resubmission period as an excuse.   

You may check if you could retrieve the gene expression. 

https://doi.org/10.1016/j.molp.2019.06.006
http://dx.doi.org/10.1016/j.molp.2017.01.004  

These data would support the in silico predictions. At least the authors give the gene expression data either tracing in previously reported transcriptome datasets or, if possible, from their own experiment. 

The prediction data is lame without any evidence of biological significance; the authors would agree with me on this. 

Reviewer 3 Report

The authors responded to all the questions raised in the review early. However, there could be two reasons, which holds the major concern to warrants the publication in Plants. a) The study was conducted in the model species, it would be great, this type of study could be conducted in the non-model plant species, if possible, to any food crops. b) Since, the author claims it is the first to report on the pollen-specific, there were studies on the protein sequences, so again, the current study carries an average novelty, more importantly, validation of predicted sequence using at least a tobacco infiltration assay would add weightage. so, I would like to leave the final decision to the editors.